# A Biologically Interpretable Graph Convolutional Network to Link Genetic Risk Pathways and Neuroimaging Markers of Disease

Sayan Ghosal[1], Qiang Chen[2], Giulio Pergola[2,3], Aaron L. Goldman[2], William Ulrich[2], Daniel R. Weinberger[2], and Archana Venkataraman[1]

[1]Department of Electrical and Computer Engineering, Johns Hopkins University, USA
[2]Lieber Institute for Brain Development, USA
[3]Department of Basic Medical Sciences, University of Bari Aldo Moro, Italy

## Abstract

We propose a novel end-to-end framework for whole-brain and whole-genome imaging-genetics. Our genetics network uses hierarchical graph convolution and pooling operations to embed subject-level data onto a low-dimensional latent space. The hierarchical network implicitly tracks the convergence of genetic risk across well-established biological pathways, while an attention mechanism automatically identifies the salient edges of this network at the subject level. In parallel, our imaging network projects multimodal data onto a set of latent embeddings. For interpretability, we implement a Bayesian feature selection strategy to extract the discriminative imaging biomarkers; these feature weights are optimized alongside the other model parameters. We couple the imaging and genetic embeddings with a predictor network, to ensure that the learned representations are linked to phenotype. We evaluate our framework on a schizophrenia dataset that includes two functional MRI paradigms and gene scores derived from Single Nucleotide Polymorphism data. Using repeated 10-fold cross-validation, we show that our imaging-genetics fusion achieves the better classification performance than state-of-the-art baselines. In an exploratory analysis, we further show that the biomarkers identified by our model are reproducible and closely associated with deficits in schizophrenia.

## 1 Introduction

Neuropsychiatric disorders such as autism and schizophrenia have two complementary viewpoints. On the one hand, they are associated with behavioral and cognitive dysfunction (Kay et al., 1987; Bowie et al., 2006), which results in altered brain activity (Karlsgodt et al., 2010). On the other hand, they are highly heritable, which suggests a strong genetic underpinning (Gejman et al., 2010; Ripke et al., 2014). Most studies decouple the problems of disentangling the neural mechanisms (Gur et al., 2010) and pinpointing genetic variations (Ripke et al., 2014), which ultimately provides an incomplete picture of the underlying disorder (Karlsgodt et al., 2010; Karam et al., 2010).

Imaging-genetics is an emerging field that tries to merge these complementary viewpoints (Hariri & Weinberger, 2003). The imaging features are often derived from structural and functional MRI (s/fMRI), and the genetic variants are typically captured by Single Nucleotide Polymorphisms (SNPs). Data-driven imaging-genetics methods can be grouped into four categories. The first category uses sparse multivariate regression to predict one or more imaging features based on a linear combination of SNPs (Wang et al., 2012; Liu et al., 2014). The second category uses representation learning (Pearlson et al., 2015; Hu et al., 2018) to maximally align the data distributions of the two modalities. These prior works focus on finding associations between imaging and genetics, whereas recent works are starting to incorporate additional phenotypes like disease status. The third category relies on generative models (Batmanghelich et al., 2016; Ghosal et al., 2019) to fuse imaging and genetics data with subject diagnosis. While this fusion extracts discriminative biomarkers, the generative nature makes it hard to compensate for missing data or to add data modalities. In fact, clinical neuroscience is moving towards multimodal imaging acquisitions where different modalities contain potentially orthogonal information about the disease. However, with multimodal data come the problem of missing data acquisitions, thus underscoring the need for flexible and adaptable methods.

## 2    RELATED WORK: DEEP LEARNING FOR BIOLOGICAL DATA ANALYSIS

The fourth category for imaging-genetics uses deep learning to link the multiple viewpoints (Srinivasagopalan et al., 2019; Zeng et al., 2018; Eraslan et al., 2019; Shen & Thompson, 2020). For example, the seminal work of (Ghosal et al., 2021) uses coupled autoencoder and predictor networks to integrate neuroimaging, SNPs, and diagnosis. While important, this work (and almost all imaging-genetics analyses) requires a drastically reduced set of genetic features to ensure model stability (Bhattacharjee et al., 2012). In terms of scale, a raw dataset of 100K SNPs is reduced to 1K SNPs, often based on a Genome-Wide Association Study (GWAS) (Ripke et al., 2014). In contrast, neuropsychiatric disorders are polygenetic, meaning that they are influenced by numerous genetic variants interacting across many biological pathways. The GWAS sub-selection step effectively removes the downstream information about these interactions (Horváth et al., 2015). Within the genetics realm, there is a vast literature that associates SNPs and genes to different biological pathways (Mi et al., 2013; Ashburner et al., 2000; Carbon et al., 2021). The works of (van Hilten et al., 2020; Gaudelet et al., 2020) have used this information to design a sparse artificial neural network that aggregates genetic risk according to these pathways in order to predict a phenotypic variable. While an important first step, their ANN contains just a single hidden layer, which does not account for the hierarchical and interconnected nature of the biological processes.

Graph convolutional networks (GCNs) (Kipf & Welling, 2016) provide a natural way to leverage the high-dimensional and interconnected genetic relationships. GCNs have gained immense popularity in protein structure prediction (Fout et al., 2017), drug discovery (Stokes et al., 2020), and gene-gene interactions (Yuan & Bar-Joseph, 2020). Beyond the architectural design, graph attention layers learn *edge weightings* that allows to track the information flow through the graph (Velicković et al., 2017). The attention focuses on the most discriminative set of interactions between the nodes, often leading to better generalization on unseen data. Graph attention has been successful in biological applications, like predicting disease-RNA association (Zhang et al., 2019) and essential gene prediction (Schapke et al., 2021). Finally, similar to the standard max-pooling operation, graph pooling allows us to aggregate information at each level of the hierarchy (Bruna et al., 2013; Lee et al., 2019; Ying et al., 2018). During standard graph pooling, the nodes at each level are clustered to form a smaller subgraph. These clusters can be obtained via deterministic algorithms (Bruna et al., 2013) or be learned during training (Ying et al., 2018; Lee et al., 2019). While beneficial for information compression, the clustering operation obscures the interpretation of each node. Our framework bypasses this issue by constructing a sparse hierarchical graph for convolution and pooling based on a well-known ontology.

One drawback of deep learning is the black-box nature of the models. Recently, Lundberg et al. (2017) has provided a framework, SHAP (SHapley Additive exPlanations), to compute feature importance for interpretability. SHAP assigns a score to each feature equal to the marginal contribution of that feature to the output, as averaged over all possible feature subsets. Variants like kernel-SHAP provide for increased flexibility of the feature importance computation (Ribeiro et al., 2016). While SHAP provides a model-agnostic tool to quantify input-output relationships, it says little about how groups of input features affect the output, which is a primary interest in biological datasets.

In this paper, we introduce an interpretable **G**enetics and m**U**ltimodal Imaging based **DE**ep neural network (**GUIDE**), for whole-brain and whole-genome analysis. We use hierarchical graph convolutions that mimic an ontology based network of biological processes (Mi et al., 2013). We use pooling to capture the information flow through the graph and graph attention to learn the salient edges, thus identifying key *pathways* associated with the disorder. On the imaging front, we use an encoder coupled with Bayesian feature selection to learn multivariate importance scores. The latent embeddings learned by our genetics graph convolutions and imaging encoder are combined for disease prediction. We demonstrate that both the ontology network and the imaging-genetics fusion in GUIDE achieve better classification accuracy than state-of-the-art baselines. More importantly, GUIDE can identify robust and clinically relevant biomarkers in both data domains.

## 3    A DEEP MULTIMODAL IMAGING-GENETICS FRAMEWORK

Fig. 1 illustrates our GUIDE framework. Inputs are the gene scores $\mathbf{g}_n \in \mathbf{R}^{G \times 1}$ for each subject $n$ and the corresponding imaging features $\mathbf{i}_n^1 \in \mathbf{R}^{M_1 \times 1}$ and $\mathbf{i}_n^2 \in \mathbf{R}^{M_2 \times 1}$ obtained from two different acquisitions. The gene scores $\mathbf{g}_n$ are obtained by grouping the original SNPs according to the nearest

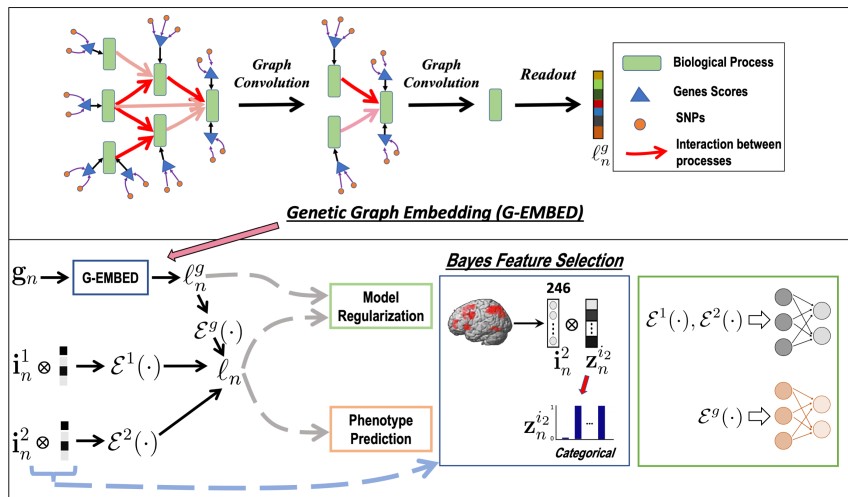

Figure 1: Overview of the GUIDE framework. **Top:** Gene embedding using attention based hierarchical graph convolution. **Bottom:** Imaging and genetics integration; both modalities are coupled for phenotype prediction—in our case, disease classification. The variables $\{\mathbf{i}_n^1, \mathbf{i}_n^2\}$ correspond to the imaging data, and $\mathbf{g}_n$ is the genetic data. $\mathcal{E}(\cdot)$ is a set of fully connected layers that encodes the input data modalities. We further provide the architectural details of our model in Appendix A.1

gene and aggregating the associated genetic risk weighted by GWAS effect size (Ripke et al., 2014). The subject diagnosis (phenotype) $y_n \in \{0, 1\}$ is known during training but is absent during testing.

## 3.1 GENETIC GRAPH EMBEDDING

The top portion of Fig. 1 illustrates our attention-based graph convolution model for genetic embedding. The underlying graph is based on the gene ontological hierarchy (Ashburner et al., 2000) that effectively maps each gene to different biological processes. Mathematically, this ontology gives rise to a sparse matrix $\mathbf{A}_g \in \{0, 1\}^{G \times T}$, where $G$ is the total number of genes, and $T$ is the number of biological processes (i.e., nodes). Given this substrate, GUIDE first learns a projection of the gene scores $\mathbf{g}_n$ onto $T$ graph nodes. Each node signal is a $d$-dimensional feature vector, $\mathbf{p}_n(t) \in \mathbf{R}^{1 \times d}$.

$$\mathbf{p}_n(t) = PReLU(\mathbf{g}_n^T (\mathbf{W}_t \otimes \mathbf{A}_g [:, t])),\tag{1}$$

where each of the columns of the learned weight matrix $\mathbf{W}_t \in \mathbf{R}^{G \times d}$ are masked by the nonzero entries in the $t^{th}$ column of $\mathbf{A}_g$. We note that this projection step is similar to a single-layer perceptron with only a subset of input nodes connected to each hidden node.

## 3.2 GRAPH ATTENTION AND HIERARCHICAL POOLING

In addition to grouping genes into biological processes, a standard ontology also specifies a hierarchical relationship between the biological processes themselves (Alcocer-Cuarón et al., 2014). An example of this hierarchy is: generation of neurons → neurogenesis → nervous system development.

We use a graph convolution to mimic the flow of information through the hierarchy of biological processes. Here, GUIDE learns two complementary pieces of information. The first is a set of graph convolutional filters that act on each embedded node signal, and the second is a set of subject-specific attention weights that select the discriminative edges. Formally, let the binary matrix $\hat{\mathbf{A}}_p \in \mathbf{R}^{T \times T}$ capture the directed edges in the ontology. Our graph convolution at stage $l$ is:

$$\mathbf{p}_n^{l+1}(t) = \sigma \left( \sum_{j \in Child(t)} \mathbf{E}_n^l(t, j) \mathbf{p}_n^l(j) \mathbf{W}^l + \beta_t \mathbf{p}_n^l(t) \mathbf{W}_s \right),\tag{2}$$

where $\mathbf{p}_n^l(t) \in \mathbf{R}^{1 \times d_l}$ is signal for node $t$ at stage $l$, $\mathbf{W}^l \in \mathbf{R}^{d_l \times d_{l+1}}$ is the convolutional filter between stages $l$ and $l + 1$, $\beta_t$ is the self-influence for node $t$, $\mathbf{W}_s \in \mathbf{R}^{d_l \times d_{l+1}}$ is the convolution

filter for self loop, and $\sigma(\cdot)$ is the nonlinearity. The summation in Eq. (2) aggregates the influence over all child nodes defined by the graph $\mathbf{A}_p$, thus respecting the high-level ontology.

The variables $\mathbf{E}_n^l(t, j)$ quantify the influence of the child node $j$ over the parent node $t$ at convolutional stage $l$. Unlike a standard graph convolutional network, in which the edge weights are fixed, we learn $\mathbf{E}_n^l(t, j)$ using a graph attention mechanism. Mathematically, we have

$$\mathbf{E}_n^l(t, j) = \frac{\exp\left(\tanh\left(\left[\mathbf{p}_n^l(t)\mathbf{W}^l \quad \mathbf{p}_n^l(j)\mathbf{W}^l\right] \cdot \mathbf{c}^l\right)\right)}{\sum_{j \in Child(t)} \exp\left(\tanh\left(\left[\mathbf{p}_n^l(t)\mathbf{W}^l \quad \mathbf{p}_n^l(j)\mathbf{W}^l\right] \cdot \mathbf{c}^l\right)\right)} \tag{3}$$

where $\mathbf{c}^l$ is a fixed weight vector learned during training. We estimate the self influence variable as $\beta_t = \sigma\left(\mathbf{p}_n^l(t)\mathbf{W}_s \cdot \mathbf{c}_s^l\right)$, where $\mathbf{c}_s^l$ is another weight vector learned during training.

Finally, we use hierarchical pooling to coarsen the graph. As formulated in Eq. (2), the graph convolution passes information "upwards" from child nodes to parent nodes. From the ontology standpoint, each stage of the hierarchy goes from a lower-level biological process (e.g., neurogenesis) to a higher-level biological process (e.g., nervous system development). Thus, we remove the lowest (leaf) layer from the graph at each stage $l$ and continue this process until we reach the root nodes. The genetic embedding $\ell_n^g \in \mathbf{R}^{\mathcal{D} \times 1}$ is obtained by concatenating the signals from each root node.

### 3.3 BAYESIAN FEATURE SELECTION

The bottom branch of Fig. 1 shows our embedding and feature selection procedure for the imaging modalities. Our dataset in this work contains two fMRI paradigms, leading to inputs $\mathbf{i}_n^1$ and $\mathbf{i}_n^2$ of each subject $n$. However, our framework naturally extends to an arbitrary number of modalities.

From a Bayesian viewpoint, the problem of feature selection can be handled by introducing an unobserved binary random vector $\mathbf{z}_n^m$ of the same dimensionality as $\mathbf{i}_n^m$ and inferring its posterior probability distribution given the paired training dataset: $\mathcal{D} = \{\mathbf{i}_n^m, y_n\}$, where $y_n$ denotes the class label. By defining $\mathbf{I}^m = [\mathbf{i}_1^m, \ldots, \mathbf{i}_n^m]$, $\mathbf{Y} = [y_1, \ldots, y_n]$, and $\mathbf{Z}^m = [\mathbf{z}_1^m, \ldots, \mathbf{z}_n^m]$, we note that the desired posterior distribution $p(\mathbf{Z}^m|\mathbf{I}^m, \mathbf{Y})$ is intractable. One strategy is to minimize the KL divergence between an approximate distribution $q(\cdot)$ and the true posterior distribution $KL\left(q(\mathbf{Z}^m)||p(\mathbf{Z}^m|\mathbf{I}^m, \mathbf{Y})\right)$. Mathematically, this optimization can be written as

$$\underset{q(\cdot)}{\operatorname{argmin}} \quad -E_q\left[\log\left(p\left(\mathbf{Y}|\mathbf{I}^m, \mathbf{Z}^m\right)\right)\right] + KL\left(q(\mathbf{Z}^m)||p(\mathbf{Z}^m)\right), \tag{4}$$

where $p(\mathbf{Z}^m)$ is a prior over the latent masks. While Eq. (4) does not have a closed-form solution, it can be optimized via Monte Carlo integration by sampling the vectors $\mathbf{z}_n^m \sim Bernoulli(\mathbf{b}^m)$, where $\mathbf{b}^m$ parameterizes the approximate distribution $q(\cdot)$, and minimizing the empirical form of Eq. (4) (Gal & Kendall; Kingma & Welling). In this case, the first term becomes the binary cross entropy loss where the input features $\mathbf{i}_n^m$ are masked according to $\mathbf{z}_n^m$. In order to learn the probability maps $\mathbf{b}^m$ during training, we replace the binary $\mathbf{z}_n^m$ with a continuous relaxation of the Bernoulli distribution:

$$\mathbf{z}_n^m = \sigma\left(\frac{\log(\mathbf{b}^m) - \log(1 - \mathbf{b}^m) + \log(\mathbf{u}_n^m) - \log(1 - \mathbf{u}_n^m)}{r}\right), \tag{5}$$

where $\mathbf{u}_{n_i}^m$ is sampled from $Uniform(0, 1)$, the parameter $r$ controls the relaxation from the $\{0, 1\}$ Bernoulli, and the feature selection probabilities $\mathbf{b}^m$ are learned during training (see Section 3.4).

### 3.4 MULTIMODAL FUSION AND MODEL REGULARIZATION

As shown in Fig. 1, the Bayesian feature selection step is followed by a cascade of fully connected layers, denoted $\mathcal{E}^m(\cdot)$, to project each imaging modality $m$ onto a low-dimensional latent embedding. Likewise, the genetic embedding $\ell_n^g$ is passed through a separate fully connected cascade $\mathcal{E}^g(\cdot)$ and onto the same low-dimensional space. To leverage synergies between the imaging and genetics data, we fuse the latent embeddings across modalities to obtain a common representation:

$$\ell_n = \frac{1}{M_n}\left(\mathcal{E}^1(\mathbf{i}_n^1 \otimes \mathbf{z}_n^1) + \mathcal{E}^2(\mathbf{i}_n^2 \otimes \mathbf{z}_n^2) + \mathcal{E}^g(\ell_n^g)\right), \tag{6}$$

where $\otimes$ is the Hadamard product used in the Bayesian feature selection step, $\ell_n^g = \text{G-EMBED}(\mathbf{g}_n)$, where G-EMBED$(\cdot)$ represents the genetics network based on the ontological hierarchy, and $M_n$

is the number of modalities present for subject $n$. Finally, the latent embedding $\ell_n$ is input to a classification network to tie the learned biomarkers to patient/control phenotype.

Notice that our fusion strategy encourages the latent embedding $\ell_n$ for an individual patient to have a consistent scale, even when constructed using a subset of the modalities. Thus, we can accommodate missing data during training by updating individual branches of the network based on which modalities are present. In this way, GUIDE can maximally use and learn from incomplete data.

We introduce three regularizers to stabilize the model. The genetic regularizer reconstructs the gene scores by performing a hierarchical graph decoding (G-DECODE) on $\ell_n^g$. This operation unwraps the gene encoding via the same ontology Ashburner et al. (2000). Likewise, the imaging regularizer decodes the original feature vectors from $\ell_n$ via the artificial neural networks $\mathcal{D}^m(\cdot)$. Finally, the prior over $\mathbf{z}_n^m$ appears as the KL divergence between the learned distribution $Ber(\mathbf{b}^m)$ and a binary random vector $Ber(\mathbf{s})$ with small entries to enforce sparsity. Our loss function during training is:

$$
\mathcal{L}(\mathbf{i}_1, \mathbf{i}_2, \mathbf{g}) = -\sum_{n=1}^{N} \left( y_n \log(\hat{y}_n) + (1 - y_n) \log(1 - \hat{y}_n) \right)
$$

$$
+ \sum_{m=1}^{2} KL_{\mathcal{M}} \left( Ber(\mathbf{b}^m) || Ber(\mathbf{s}) \right) + \lambda_I \sum_{n=1}^{N_1} ||\mathbf{i}_n^1 - \mathcal{D}^1(\ell_n)||_2^2 + \lambda_I \sum_{n=1}^{N_2} ||\mathbf{i}_n^2 - \mathcal{D}^2(\ell_n)||_2^2
$$

$$
+ \lambda_G \sum_{n=1}^{N_g} ||\mathbf{g}_n - \text{G-DECODE}(\ell_n^g)||_2^2 \tag{7}
$$

where $\hat{y}$ is the class prediction, $N$ is the total number of subjects, $N_m$ is the number of subjects with modality $m$ present, and the hyper-parameters $\{\lambda_I, \lambda_G\}$ control the contributions of the data reconstruction errors. The function $KL_{\mathcal{M}}(\cdot || \cdot)$ in Eq. (7) averages the element-wise KL divergences across the input feature dimension, thus maintaining the scale of the prior term regardless of dimensionality.

The first two terms of Eq. (7) correspond to the classification task and the feature sparsity penalty, which are empirical translations of Eq. (4). The final three terms are the reconstruction losses, which act as regularizers to ensure that the latent embedding is capturing the original data distribution.

**Training Strategy:** As described in Section 4, our dataset consists of 1848 subjects with only genetics data and an additional 208 subjects with both imaging and genetics data. Given the high genetics dimensionality, we pretrain the G-EMBED and G-DECODE branches and classifier of GUIDE using the 1848 genetics-only subjects. These 1848 subjects are divided into a training and validation set, the latter of which is used for early stopping. We use the pretrained model to warm start GUIDE framework and perform 10-fold nested CV over the 208 imaging-genetics cohort.

We learn the Bayesian feature selection probabilities during training by sampling the random vectors $\mathbf{z}_n^1, \mathbf{z}_n^2$ during each forward pass using Eq. (5) and using them to mask the inputs $\mathbf{i}_n^1, \mathbf{i}_n^2$ for patient $n$. Finally, if a data modality is missing for subject $n$, we simply fix the corresponding encoder-decoder branch and update the remaining branches and predictor network using backpropagation.

**Implementation Details:** We use the first five layers of the gene ontology (Ashburner et al., 2000) to construct G-ENCODE. In total, this network encompasses 13595 biological processes organized from 2836 leaf nodes to 3276 root nodes. We perform a grid search over three order of magnitude and fix the hyperparameters $\lambda_I = 3 \times 10^{-3}$, $\lambda_G = 10^{-5}$. We fix the signal dimensionality at $d = 2$ for the gene embedding and $d_l = 5$ for the subsequent graph convolution and deconvolution operations based on the genetics-only subjects. Likewise, the non-linearity in Eq. (2) selected to be a $LayerNorm$, followed by $PReLU$ and $Dropout$, once again using the genetics-only subjects. The Bernoulli prior over $\mathbf{z}^m$ is set at $s = 0.001$, which is consistent with Ghosal et al. (2021). Further architectural details are provided in the Appendix. We train GUIDE using ADAM with an initial learning rate of 0.0002 and decay of 0.7 every 50 epochs. Our code is implemented using Matlab 2019b and PyTorch 3.7. Training the full model takes roughly 17hrs on a 4.9GB Nvidia K80 GPU.

## 3.5 BASELINE COMPARISON METHODS

We compare GUIDE with two conventional imaging-genetics methods and with single modality versions of our framework. In each case, the hyperparameters are optimized using a grid search.

**Parallel ICA + RF:**  We concatenate the imaging modalities to single vector $\mathbf{i}_n = [\mathbf{i}_n^{1^T} \quad \mathbf{i}_n^{2^T}]^T$ and perform parallel ICA (p-ICA) (Pearlson et al., 2015) with the gene scores $\mathbf{g}_n$. Since p-ICA cannot handle missing modalities, we fit a multivariate regression model to impute a missing imaging modality from the available ones. Specifically, if $\mathbf{i}_1^n$ is absent, we impute it as: $\mathbf{i}_1^n = \boldsymbol{\beta}\,\mathbf{i}_2^n$, where $\boldsymbol{\beta}$ is the regression coefficient matrix obtained from training data. After imputation, we use p-ICA to decompose the imaging and genetics data into independent but interrelated networks:

$$\mathbf{i}_n = \mathbf{S}\,\mathbf{e}_n \quad \text{and} \quad \mathbf{g_n} = \mathbf{W}\,\mathbf{f}_n$$

where $\mathbf{S}, \mathbf{W}$ are independent source matrices and the $\mathbf{e}_n, \mathbf{f}_n$ are loading vectors. We concatenate the loading matrices $\left[\mathbf{e}_n^T, \mathbf{f}_n^T\right]$ and use it as the input feature vector for a random forest classifier.

During training, we apply p-ICA to just the training data to estimate the sources $\{\mathbf{S}_{train}, \mathbf{W}_{train}\}$. We use these estimated sources to obtain the loading matrices for the test data as follows:

$$\mathbf{i}_{test} = \mathbf{S}_{train}\mathbf{e} \quad \text{and} \quad \mathbf{g}_{test} = \mathbf{W}_{train}\mathbf{f}$$

**G-MIND:**  The G-MIND architecture by Ghosal et al. (2021) is designed for a 1242 genetics input. Thus, we introduce a fully-connected layer to project the high-dimensional gene scores $\mathbf{g}_n$ onto a 1242 dimensional vector for input to G-MIND (Ghosal et al., 2021). We evaluate both random weight initialization and pretraining the genetics branch of G-MIND with the 1848 genetics-only subjects.

**Single Modality Prediction:**  We consider two versions of GUIDE. The first consists of the genetics branches and classifier, and the second consists of just the imaging branches and classifier. We optimize these networks architectures using repeated 10-fold cross validation outlined in Section 3.4.

## 3.6   EVALUATION STRATEGY

We conduct a comprehensive evaluation of our framework that includes influence of embedding *a priori* biological information, biomarker reproducibility, and classification performance.

**Influence of the Ontology-Based Hierarchy:**  In GUIDE we assume that processing the data according to an established gene ontology will extract more robust and discriminative biomarkers. To test this assumption, we compare our gene ontology network to random networks and to an unstructured model. The random networks contain the same number of nodes in each layer as the ontology-based graph. However, we randomly permute edges between the parents and children nodes, and between the genetic inputs $\mathbf{g}_n$ and nodes. The corresponding unstructured model is a fully-connected ANN with the same number of parameters, but no inherent structure between layers. As a benchmark, we compare the deep learning models with the gold-standard Polygenetic Risk Score (PRS) for schizophrenia developed by the PGC consortium (Ripke et al., 2014). Broadly, the PRS is a weighted combination of the risk alleles for schizophrenia, as determined by a large GWAS study. We run a logistic regression on the (scalar) PRS to determine class membership.

**Classification Performance:**  We use repeated 10-fold cross validation (CV) on the 208 imaging-genetics subjects to quantify the performance. The models are trained using 8 folds, and the remaining two folds are reserved for validation and testing. We report accuracy, sensitivity, specificity, Area Under the ROC Curve (AUROC), and Area Under the precision-recall Curve (AUPRC). The operating point is chosen by minimizing the classification error on the validation set. We further use DeLong tests to compare statistical difference in AUROC between GUIDE and the baselines.

**Reproducibility of Feature Importance Maps:**  The probability maps $\mathbf{b}_m$ capture the importance of each feature of modality $m$. We evaluate both the predictive performance and the reproducibility of our Bayesian feature selection (BFS) scheme. We extract the the top-$K$ features of $\mathbf{b}_m$ learned during each training fold of our repeated 10-fold CV setup and encode this information as a binary indicator vector, where the entry '1' indicates that the feature is among the top-$K$. We compare the BFS features with Kernel SHAP (Lundberg et al., 2017). The background values for K-SHAP are fixed to the average input data across the training set. We use the validation set to select the top $K$ K-SHAP features and encode them in an binary indicator vector.

To quantify predictive performance, we mask our test data from each fold using the BFS and K-SHAP indicator vectors and send it through GUIDE for patient versus control classification. To quantify

the reproducibility of the top-$K$ features, we calculate the pairwise $cosine$ similarity between all the binary vectors across the folds as identified by either BFS or K-SHAP. The distribution of similarities tells us how often the same imaging features are selected across subsets of the data.

**Discovering of Biological Pathways:** The attention layer of G-ENCODE provides a subject-specific measure of "information flow" through the network. To identify discriminative pathways, we first trace all possible paths between leaf and root nodes. The importance of edges along these paths are used in a logistic regression framework to predict the subject class label. We then perform a likelihood ratio test and obtain a p-value (FDR corrected) for each path in terms of patient/control differentiation. For robustness, we repeat this experiment 10 times using subsets of $90\%$ of our total dataset $1848 + 208$ subjects and select pathways that achieve $p < 0.05$ in at least 7 of the 10 subsets.

## 4    RESULTS

**Data and Preprocessing:** We evaluate GUIDE on a study of schizophrenia provided by Anonymous Institution that contains SNP data and two fMRI paradigms.

Illumina Bead Chipset including 510K/ 610K/660K/2.5M is used for genotyping. Quality control and imputation were performed using PLINK and IMPUTE2. The resulting 102K linkage disequilibrium independent ($r^2 < 0.1$) indexed SNPs are grouped to the nearest gene (within 50kb basepairs) (Wong et al., 2017). The 13,908 dimensional input genes cores are computed as weighted average of the SNPs using GWAS effect size (Ripke et al., 2014). We note that the GWAS was performed on a separate dataset that did not include our site.

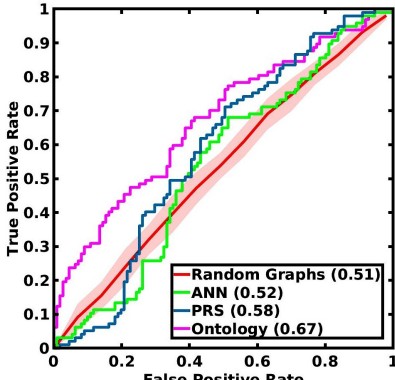

Figure 2: ROCs for the PRS (blue), unstructured ANN (green) and the structured models where G-EMBED and G-DECODE use either random graphs (red) or the gene ontology network (magenta). The AUROC is given in parentheses.

Imaging data include two task-fMRI paradigms commonly used in the schizophrenia literature. The first paradigm is a working memory task (N-back), and the second is a simple declarative memory task (SDMT). FMRI data was acquired on 3T GE Sigma scanner (EPI, TR/TE $= 2000/28$ msec; res $= 3.75 \times 3.75 \times 6/5$ for N-back and SDMT, respectively). FMRI preprocessing includes slice timing correction, realignment, spatial normalization to MNI template, smoothing and motion regression. We use SPM 12 to generate contrast maps and the Brainnetome atlas (Fan et al., 2016) to define 246 cortical and subcortical regions. The inputs $\mathbf{i}_n^1, \mathbf{i}_n^2$ correspond to the average contrast over voxels in the respective region. Additional details are reported in Appendix A.2

In total, our dataset contains $1848$ subjects (792 schizophrenia, 1056 control) with just SNP data and 208 subjects with imaging and SNP data, divided as follows: 98 subjects (42 schizophrenia, 56 control) with SNP and N-back data, 48 subjects (17 schizophrenia, 31 control) with SNP and SDMT data, and 62 subjects (38 schizophrenia, 24 control) who have all the three data modalities.

**Benefit of the Gene Ontology Network:** To quantify the value of embedding a biological hierarchy into our model architecture, we first train just the genetics branch and classifier for all three deep networks (unstructured, random graphs, ontology) using the $1848$ genetics-only subjects and test them on the SNP data from the other 208 subjects. We sample 10 random graphs in this experiment.

Fig. 2 illustrates the ROC curve over the test data. As seen, the testing performance is near-chance using both random graphs and the ANN. The gold-standard PRS does slightly better; however, this measure was derived on a much larger consortium dataset. In contrast, our gene ontology network (magenta line in Fig. 2) achieves the best performance of any method, suggesting that our biologically-inspired architecture can extract robust and predictive genetic biomarkers of a disorder.

**Classification Performance:** Table 1 reports the 10-fold CV testing performance of all the methods on the multimodal imaging-genetics dataset; we repeat the CV procedure 10 times to obtain standard deviations for each metric. We note that P-ICA and G-MIND with random initialization have relatively poor performance, likely due to the high dimensionality of $\mathbf{g}_n$ and low sample size. Pretraining on

Table 1: Classification performance (mean ± std) across repeated CV runs. P-values obtained from DeLong test indicate significantly greater AUROC for GUIDE than each of the baselines.

| Perf / Method | Sensitivity | Specificity | Accuracy | AUPRC | AUROC | P-Value |
|---|---|---|---|---|---|---|
| P-ICA | $0.30 \pm 0.10$ | $\mathbf{0.80 \pm 0.07}$ | $0.56 \pm 0.03$ | $0.54 \pm 0.04$ | $0.59 \pm 0.04$ | $< 10^{-4}$ |
| G-MIND (random) | $\mathbf{0.62 \pm 0.06}$ | $0.65 \pm 0.05$ | $0.63 \pm 0.02$ | $0.62 \pm 0.03$ | $0.67 \pm 0.03$ | $< 10^{-4}$ |
| G-MIND (pretrain) | $0.60 \pm 0.07$ | $0.66 \pm 0.07$ | $0.63 \pm 0.03$ | $0.62 \pm 0.03$ | $0.68 \pm 0.02$ | $< 10^{-4}$ |
| Imaging Only | $0.44 \pm 0.18$ | $0.76 \pm 0.14$ | $0.61 \pm 0.01$ | $0.62 \pm 0.02$ | $0.66 \pm 0.02$ | $< 10^{-4}$ |
| Genetic Only | $0.54 \pm 0.15$ | $0.69 \pm 0.10$ | $0.62 \pm 0.02$ | $0.63 \pm 0.03$ | $0.68 \pm 0.02$ | $< 10^{-4}$ |
| GUIDE (Random Dropout) | $0.51 \pm 0.14$ | $0.79 \pm 0.12$ | $0.66 \pm 0.02$ | $\mathbf{0.70 \pm 0.02}$ | $\mathbf{0.75 \pm 0.01}$ | $0.27$ |
| GUIDE | $\mathbf{0.62 \pm 0.04}$ | $0.76 \pm 0.04$ | $\mathbf{0.69 \pm 0.01}$ | $\mathbf{0.70 \pm 0.03}$ | $\mathbf{0.75 \pm 0.01}$ | |

a separate genetics dataset improves the AUC of G-MIND, highlighting the benefits of increased data. We also evaluate replacing the GUIDE BFS layer with random dropout. While the quantitative performance is similar, the feature reproduciblity is much higher with BFS (see Appendix A.5). This result underscores the dual value of BFS for predictive performance and biomarker discovery.

**Reproducibility of BFS Features:** Fig. 3 illustrates the classification AUC when the input features are masked according to the top-$K$ importance scores learned by the BFS (solid blue) and K-SHAP (dashed red) procedures. The confidence intervals are obtained across the repeated CV folds. As seen, both feature selection schemes achieve similar AUCs as the number of features $K$ is varied across its entire range, thus highlighting the robustness of our (simpler) BFS approach.

Fig. 4 illustrates the distribution of $cosine$ similarities between the masked feature vectors learned by BFS and K-SHAP for $K = 50$. Once again, the repeated CV procedure is run 10 times, yielding 100 total folds and 4950 pairwise comparisons per method. Notice that our BFS procedure achieves significantly higher $cosine$ similarity values, which suggests that it selects a more robust set of features that is consistent across subsets of our main cohort. In contrast, K-SHAP relies heavily on individual subject data, which varies considerably across random subsets of the cohort.

**Imaging Biomarkers:** Fig. 5 illustrates the consistent imaging features that are selected by our method across the folds for $K = 50$. We have colored each brain region according to the selection frequency. For clarity, we have displayed only the top $40\%$ regions. We observe that the N-back biomarkers involve the Middle Frontal Gyrus (MFG), Inferior Frontal Gyrus (IFG), and default mode network which are associated with schizophrenia (Callicott et al., 2003; Wang et al., 2015). The SDMT biomarkers implicate the Supramarginal Gyrus (SMG), Superior Frontal Gyrus (SFG), along with Precuneus and Cuneus. We further interpret the higher order brain states implicated by these regions using Neurosynth (Yarkoni et al., 2011). Neurosynth uses a reverse-inference procedure to select a set of "cognitive terms" associated with a set of input coordinates based on how frequently similar patterns have been observed across the fMRI literature. Fig. 5 shows that the N-back and SDMT biomarkers are associated with memory retrieval and attention (Carter et al., 2010; Guo et al., 2019), thus verifying that GUIDE captures information relevant to the fMRI tasks.

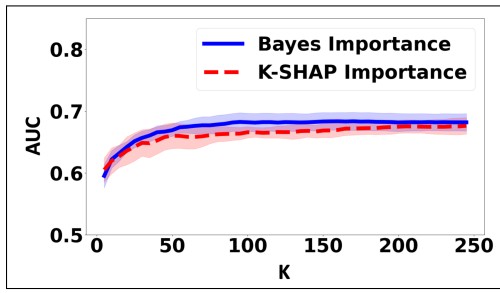

Figure 3: Mean AUC and confidence interval when masking the top-$K$ imaging features learned by BFS (solid blue) and K-SHAP (dashed red). $K$ is varied along the x-axis.

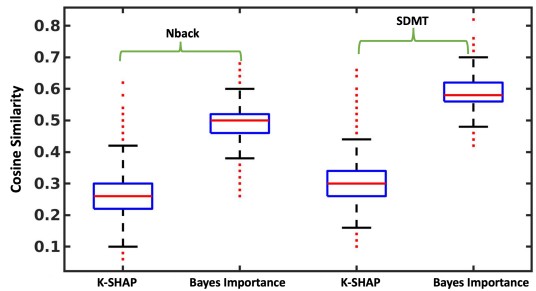

Figure 4: Distribution of pairwise $cosine$ similarities between the feature masks learned by BFS and K-SHAP across the repeated CV folds. Here, we fix $K = 50$ for both.

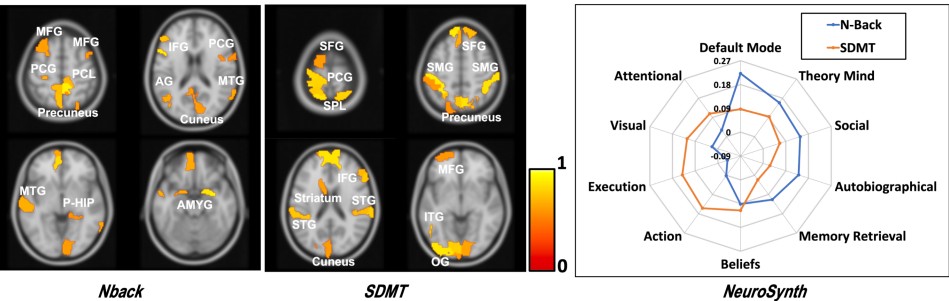

Figure 5: **Left:** The consistent set of brain regions captured by the dropout probabilities $\{\mathbf{b}^1, \mathbf{b}^2\}$ for $K = 50$. The color bar denotes the selection frequency. **Right:** Brain states associated with the selected regions for each fMRI task, as decoded by Neurosynth.

**Genetic Pathways:** GUIDE contains $14881$ "biological pathways" between the leaf and root nodes. Using the likelihood ratio test outlined in Section 3.6, we are able to identify $152$ pathways with $p < 0.05$ after FDR correction that appear in at least 7 of the 10 random training datasets. We cluster these pathways into $10$ categories based on their semantic similarity. Here, we use the tf-idf (Beel et al., 2016) information retrieval scheme to extract keywords in the pathways; we then embed them in a two-dimensional space using t-SNE (Van Der Maaten et al., 2008) and apply a k-means clustering algorithm. Fig. 6 shows the clusters along with the most frequent keywords within each cluster. As seen, the frequent biological processes involve calcium signaling, regulation of macrophage and immunological synapse formation which have been previously linked to schizophrenia (Ormel et al., 2017; Berridge, 2013; Van Kesteren et al., 2017). This exploratory experiment shows that GUIDE can be used to extract discriminative biological information about neuropsychiatric disorders.

## 5 CONCLUSION

We propose a novel biologically interpretable graph convolutional network that integrates imaging and genetics data to perform disease prediction. This model is able to leverage prior biological information of different connected biological processes to identify patterns from both imaging and genetic data sets. Additionally, the unique use of Bayesian feature selection is able to find a set of clinically relevant biomarkers. The improved classification performance shows the ability of this model to build a comprehensive view about the disorder based on the incomplete information obtained from different modalities. We note that, our framework can easily be applied to other imaging modalities, such as structural and diffusion MRI, simply by adding encoder-decoder branches. In future, we will apply our framework to other application domains, such as autism and Parkinson's disease.

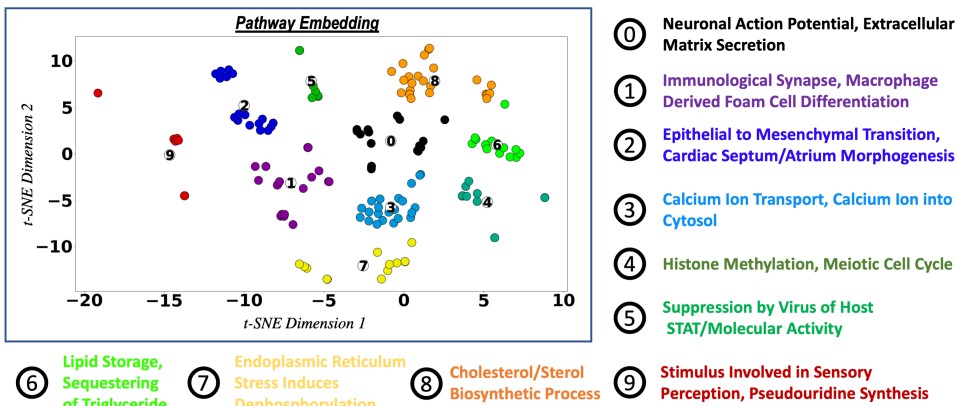

Figure 6: Ten different categories of pathways based on their semantic similarity. The key words show the most frequent biological processes within each cluster.

**Acknowledgements:** This work was supported by NSF CRCNS 1822575, NSF CAREER 1845430, and the National Institute of Mental Health extramural research program.

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

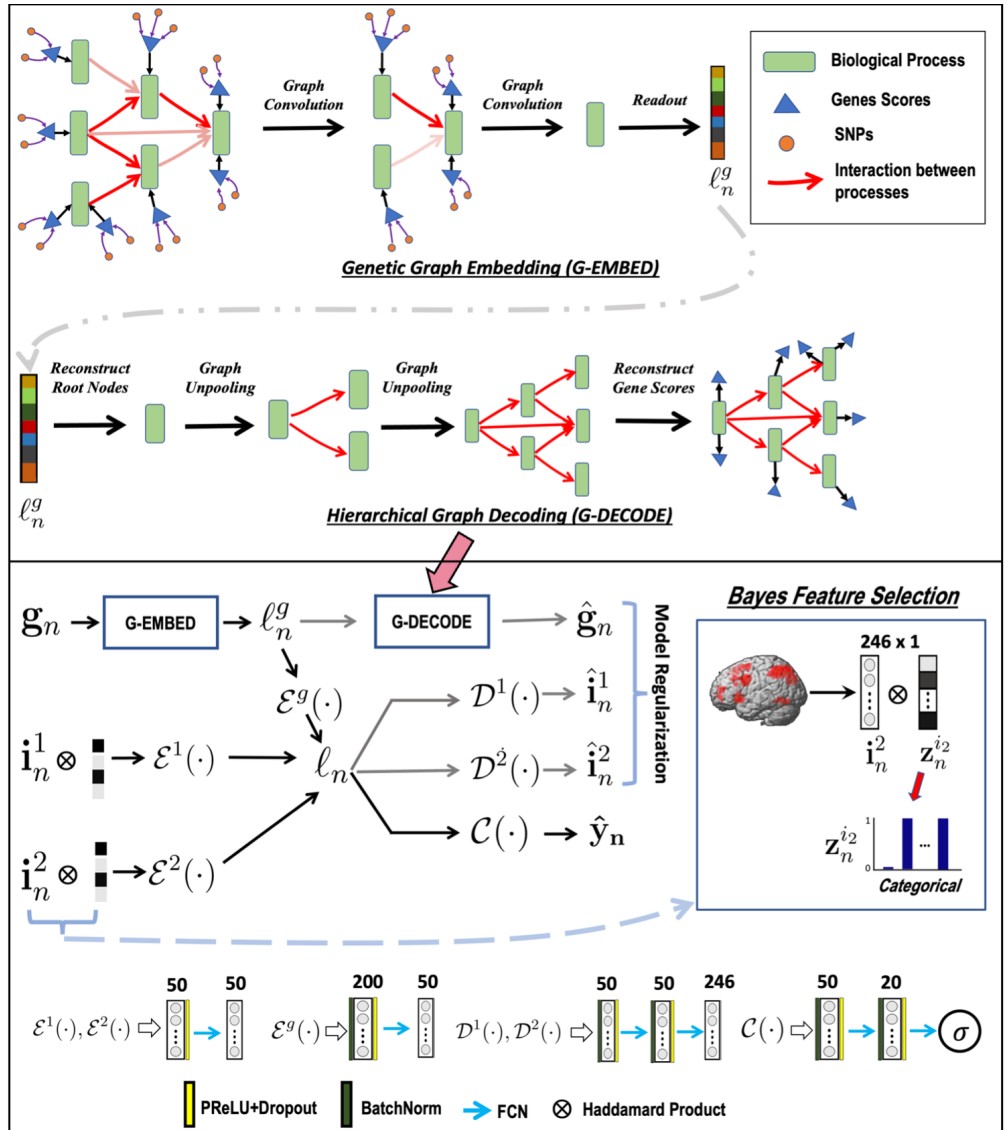

Figure 7: Overview of the GUIDE framework. **Top:** Gene embedding using attention based hierarchical graph convolution. We also depict the unpooling operation used as a regularizer. **Bottom:** Imaging and genetics integration; both modalities are coupled for disease classification. The variables $\{\mathbf{i}_n^1, \mathbf{i}_n^2\}$ correspond to the imaging data, and $\mathbf{g}_n$ is the genetic data. $\mathcal{E}(\cdot), \mathcal{D}(\cdot), \mathcal{C}(\cdot)$ are the feature extraction, model regularization, and classification operations, respectively.

## A  APPENDIX

### A.1  DETAILED GUIDE ARCHITECTURE

GUIDE is a biologically interpretable deep neural network that couples imaging and genetics data for simultaneous biomarker discovery and phenotypic prediction. GUIDE consists of a graph-based genetic encoder, a parallel imaging encoder, a phenotypic predictor, and model regularizers. The genetic module uses hierarchical graph convolutions and pooling operations to strategically embed the genetic data to latent space. This architecture mimics the organization of a well-established gene ontology. The imaging encoder uses fully connected layers to project the data to the latent space. Finally, the predictor uses the fused latent representation for disease diagnosis. The regularizers reconstruct the data to ensure that the latent embedding is capturing the original data distribution.

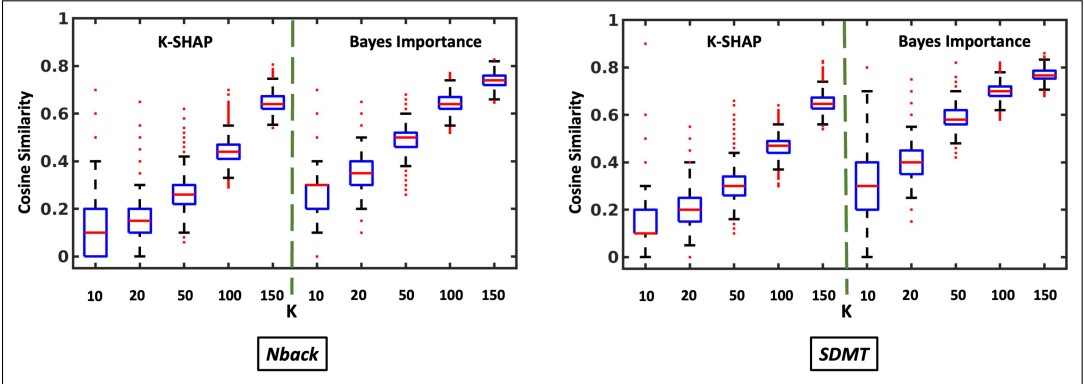

Figure 8: The reproducibility of imaging biomarkers when the feature selection has been done using K-SHAP vs Bayesian dropout. **Left** shows the performance on Nback data, **Right** shows the performance on SDMT data

Fig. 7 illustrates the detailed architecture of GUIDE. The top portion captures the genetic encoding and the decoding process. We use the gene ontology network to build the graph and use hierarchical pooling to control the flow of information according to the relationships between the biological processes. At the same time, an graph attention mechanism tracks the salient edges at the subject level. The bottom portion shows the integration of the imaging and genetics modules. G-EMBED, and $\mathcal{E}^m(\cdot)$ encodes the genetics and imaging data, respectively, and fuse the latent representation to $\ell_n$. The latent embedding is passed through decoding operations (G-DECODE, $\mathcal{D}^m(\cdot)$) to reconstruct the input data. These decoding operations act as regularizers and stabilize the model.

### A.1.1 Further Validation of the Bayesian Feature Selection

The Bayesian feature selection layer of GUIDE is used to learn discriminative imaging biomarkers for phenotypic prediction, as captured by the probability maps $\mathbf{b}_m$. In this section, we evaluate the stability of the biomarkers across different subsets of the cohort. To start, we extract the top-$K$ features of $\mathbf{b}_m$ learned during each training fold of our repeated 10-fold CV setup and encode this information as a binary indicator vector, where '1' indicates that the feature is among the top-$K$. We compare the Bayesian Feature Selection ((BFS) with Kernel SHAP (Lundberg et al., 2017). To quantify the reproducibility of the top-$K$ features, we calculate the pairwise *cosine* similarity between all the binary vectors across the folds as identified by either BFS or K-SHAP. The distribution of similarities tells us how often the same imaging features are selected across subsets of the data.

Fig. 8 reports the distribution of *cosine* similarities between the masked feature vectors learned by BFS and K-SHAP for $K = 10, 20, 50, 100, 150$. The repeated CV procedure is run 10 times, yielding 100 total folds and 4950 pairwise comparisons per method. Notice that our BFS procedure achieves significantly higher *cosine* similarity values at each setting for $K$, which suggests that it selects a more robust set of features that is consistent across subsets of our main cohort.

### A.2 Data Description

We validate our framework on fMRI and genetic data acquired at the Anonymous Institution. Our imaging data include two standard task-fMRI paradigms acquired from 208 subjects. This section provides the details of the experimental paradigms and the subject demographics.

**Neuroimaging Data:** As shown in Fig. 9, our data include two standard task-fMRI paradigms that were previously designed to study memory deficits in schizophrenia (*Anonymous et al., Anonymous et al.*). The first paradigm is a block design working memory task (N-Back). During the 0-back blocks, participants were instructed to press a button corresponding to a number displayed on the screen. During the 2-back working memory blocks, participants were instructed to recall the number they had seen two stimuli previously. We use Generalized Linear Model (GLM) to obtain the activation map for each task block. The final contrast is the subtraction $\beta_{2-back} - \beta_{0-back}$. We compute the region-

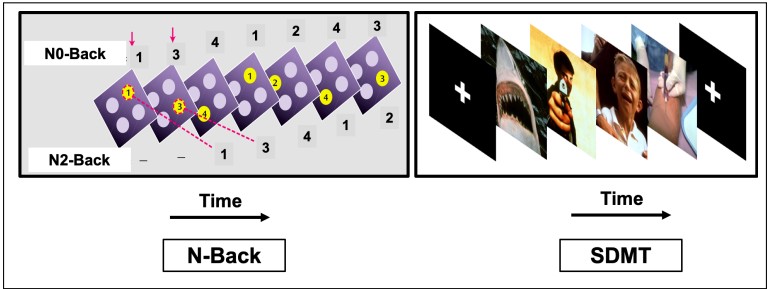

Figure 9: **Left** The experimental paradigm of the N-Back task. The top row shows a sample response for N0-Back and the bottom row shows a sample response for N2-Back. **Right** The experimental setup for the SDMT task. Subjects are asked to view neutral and aversive stimuli.

| Demographic | Anonymous | |
| --- | --- | --- |
| | N-back | SDMT |
| Sex (M/F) | 113/47 | 70/40 |
| Age (years) | $31 \pm 10$ | $31 \pm 10$ |
| Education (years) | $15 \pm 2$ | $12 \pm 2$ |
| IQ | $105 \pm 10$ | $104 \pm 10$ |

Table 2: Demographic information for subjects provided by Anonymous Institution.

| Data Sets | | | | | | | |
| --- | --- | --- | --- | --- | --- | --- | --- |
| SNP-Only | | N-back + SNP | | SDMT + SNP | | All Modalities | |
| Cases | Controls | Cases | Controls | Cases | Controls | Cases | Controls |
| 793 | 1056 | 42 | 56 | 17 | 31 | 38 | 24 |

Table 3: Breakdown by patients and controls for each configuration.

wise inputs as the average of these contrast values across all voxels in each particular region. The second paradigm is a simple declarative memory task (SDMT) which involves incidental encoding of complex aversive visual scenes. Similar to N-back we estimate the brain activity associated with the encodig task. The SDMT contrast map is the subtraction $\beta_{aversive} - \beta_{crosshair}$. Again, our region wise inputs are the average of these contrast values across all voxels in each parcel of brain. Additional details for generating the contrast maps can be found in (Friston et al., 1995).

Our imaging dataset is incomplete, as many subjects were only scanned using one of the two fMRI paradigms. Table 3 reports the breakdown of patients and controls for each configuration. Finally, Table 2 reports the demographic information for the cohort. In each case, the patient and control groups were matched on age, IQ (WRAT score), and years of education.

**Gene Ontology** The ontology provides a pre-defined hierarchical network of biological processes, which has been curated by experts in biology. From a modeling perspective, the biological processes can be thought of as nodes in a graph. The two main parts of the ontology in (Ashburner et al., 2000; Mi et al., 2013) are the assignment of genes to these biological processes (i.e., the input projection layer) and the directed edges between the biological processes (i.e., the hierarchical graph).

### A.3 BASELINE HYPERPARAMETER SELECTION

We compare GUIDE with current machine learning models for imaging-genetics and with single-modality versions of our framework. We optimize the hyperparameters using grid search. This section provides the details of the hyperparameters used by the models.

**Parallel ICA + RF:** We use the standard implementation of Parallel ICA (p-ICA) from the Fusion ICA (FIT) toolbox (Rachakonda et al., 2012). The loading scores obtained from p-ICA are fed to a random forest model for classification. Our hyperparameter tuning optimizes the number and depth of the decision trees. We control the tree depth by setting the minimum number of observations per leaf node. After the grid search, these parameters are set to {No. trees $= 10000, MinleafSize = 50$}.

**GMIND:** Ghosal et al. (2021) selected the hyperparameters as powers of 10 such that the rescaled terms in the loss function lie within the same order of magnitude [1-10]. This criterion is intuitive (i.e., equal importance is given to both the imaging and genetic data), and it is not performance

Table 4: Classification performance (mean ± std) across repeated CV runs. P-values obtained from DeLong test indicate significantly greater AUROC for GUIDE than each of the ablated models.

| Perf / Method | Sensitivity | Specificity | Accuracy | AUPRC | AUROC | P-Value |
|---|---|---|---|---|---|---|
| No Attention | $0.49 \pm 0.09$ | $\mathbf{0.79 \pm 0.08}$ | $0.65 \pm 0.03$ | $0.70 \pm 0.02$ | $0.73 \pm 0.02$ | $0.01$ |
| No Feature Selection | $0.56 \pm 0.13$ | $0.73 \pm 0.10$ | $0.65 \pm 0.03$ | $0.67 \pm 0.03$ | $0.74 \pm 0.03$ | $0.28$ |
| No Decoder | $0.58 \pm 0.18$ | $0.66 \pm 0.17$ | $0.62 \pm 0.02$ | $0.65 \pm 0.02$ | $0.70 \pm 0.01$ | $< 10^{-4}$ |
| No Attention, No Feature Selection | $0.56 \pm 0.18$ | $0.72 \pm 0.13$ | $0.64 \pm 0.02$ | $0.67 \pm 0.03$ | $0.73 \pm 0.03$ | $0.05$ |
| No Attention, No Decoder | $0.50 \pm 0.16$ | $0.76 \pm 0.12$ | $0.64 \pm 0.03$ | $0.66 \pm 0.02$ | $0.71 \pm 0.02$ | $< 10^{-4}$ |
| No Feature Selection, No Decoder | $0.61 \pm 0.10$ | $0.69 \pm 0.10$ | $0.65 \pm 0.03$ | $0.63 \pm 0.03$ | $0.72 \pm 0.03$ | $0.01$ |
| No Attention, No Feature Selection, No Decoder | $0.57 \pm 0.20$ | $0.68 \pm 0.17$ | $0.63 \pm 0.03$ | $0.61 \pm 0.03$ | $0.71 \pm 0.02$ | $4 \times 10^{-4}$ |
| GUIDE | $\mathbf{0.62 \pm 0.04}$ | $\underline{0.76 \pm 0.04}$ | $\mathbf{0.69 \pm 0.01}$ | $\mathbf{0.70 \pm 0.03}$ | $\mathbf{0.75 \pm 0.01}$ | |

driven. We use a similar strategy and fix the hyperparameters of genetic reconstruction loss, imaging reconstruction loss, classification loss, and sparsity loss to $0.0001, 0.1, 1, 0.001$, respectively.

**Single Modality Prediction:** We consider two versions of GUIDE. The first consists of the genetics branches and classifier, and the second consists of just the imaging branches and classifier. These baselines probe the advantages of integrating imaging and genetics data modalities in a single framework. The hyperparameters for single modality prediction are the same as the full model.

## A.4 ABLATION STUDY

In this section, we evaluate the gain from three novel components of GUIDE: graph attention, feature selection, and the decoder for regularization. Specifically, the graph attention encourages GUIDE to focus on the discriminative interaction patterns in the genetic data, the Bayesian feature selection identifies the most predictive brain regions and the decoders ensure that the low-dimensional embedding faithfully captures the original data distribution.

Table 4 compares the performance of the ablated models with GUIDE. This ablation study allows us to quantify both the improvement of each component when they are incorporated in the model and the degradation in performance when one component is ablated from the full model. For example, removing the decoders causes the classification performance to degrade, regardless of the other components (i.e., GUIDE is better than "No Decoder" and "No Attention, No Feature Selection" is better than "No Attention, No Feature Selection, No Decoder"). We observe similar trends for both graph attention and Bayesian feature selection components. Thus, our ablation study demonstrates that all three components of GUIDE are essential for phenotypic prediction.

## A.5 ADDITIONAL BASELINE COMPARISONS

In this section, we compare the performance of GUIDE with additional baseline models.

**GUIDE (Random Dropout):** The feature selection layer of our original GUIDE framework both regularizes the model and identifies potential imaging biomarkers. In this baseline, we replace the Bayesian feature selection layer with random dropout.

**G-MIND (Sub-selection):** The G-MIND architecture proposed by (Ghosal et al., 2021) relies on a subselection of SNPs based on the p-values reported in a prior GWAS analysis of schizophrenia (Ripke et al., 2014). Here, we subselect the same set of SNPs and feed them to the G-MIND model. This baseline captures the performance change when the ontology-based representation is replaced with the GWAS sub-selection. We use the hyperparameters reported in (Ghosal et al., 2021).

**Hierarchical GCN:** GUIDE utilizes an ontology to flow the information through the graph. The gene ontology network is curated based on *a priori* information. In this baseline, we replace the

Table 5: Classification performance (mean $\pm$ std) across repeated CV runs. P-values obtained from DeLong test indicate significantly greater AUROC for GUIDE than each of the baselines

| Perf
Method | Sensitivity | Specificity | Accuracy | AUPRC | AUROC | P-Value |
|---|---|---|---|---|---|---|
| GUIDE (Random Dropout) | $0.51 \pm 0.14$ | $\mathbf{0.79 \pm 0.12}$ | $\underline{0.66 \pm 0.02}$ | $\mathbf{0.70 \pm 0.02}$ | $\mathbf{0.75 \pm 0.01}$ | $0.27$ |
| G-MIND (Sub-selection) | $\mathbf{0.63 \pm 0.08}$ | $0.67 \pm 0.06$ | $0.65 \pm 0.01$ | $0.63 \pm 0.04$ | $0.70 \pm 0.02$ | $< 10^{-4}$ |
| Hierarchical GCN | $0.48 \pm 0.17$ | $0.75 \pm 0.13$ | $0.62 \pm 0.02$ | $\underline{0.65 \pm 0.02}$ | $\underline{0.71 \pm 0.02}$ | $1.8 \times 10^{-4}$ |
| GCN+MaxPooling | $0.43 \pm 0.19$ | $\underline{0.76 \pm 0.14}$ | $0.61 \pm 0.03$ | $0.64 \pm 0.02$ | $0.69 \pm 0.02$ | $< \times 10^{-4}$ |
| GUIDE | $\underline{0.62 \pm 0.04}$ | $\underline{0.76 \pm 0.04}$ | $\mathbf{0.69 \pm 0.01}$ | $\mathbf{0.70 \pm 0.03}$ | $\mathbf{0.75 \pm 0.01}$ | |

ontology based graph with a random graph. The construction of this graph is explained in Section 3.6. This baseline captures the performance gain for embedding biological knowledge into our architecture.

**GCN + MaxPooling:** We compare our model with a standard GCN introduced by Kipf & Welling (2016). In this baseline, we generate a weighted adjacency matrix ($\mathbf{A} \in \mathbf{R}^{13908 \times 13908}$) using absolute correlation values between the gene scores from the pretraining data (i.e., 1848 genetic only subjects). This adjacency matrix acts as an undirected graph between the nodes. Unlike GUIDE, the nodes represent each gene score instead of a biological process. We also perform hierarchical pooling with a max pooling operation (Gao & Ji, 2019) to reduce the data dimension between the graph convolution layers. This is in stark contrast to GUIDE where we use biological knowledge to consolidate the information flow through the network. Here, we note that the adjacency matrix $\mathbf{A}$ is not sparse, resulting in a significantly higher computational overhead than GUIDE.

In this baseline, we use two graph convolutional layers, each followed by a max-pooling layer. The max-pooling layer reduces the number of nodes by approx $50\%$ such that after the final layer, the reduced data is of the same dimension as $\ell_g$ in GUIDE. This architecture balances the computational requirements and the data representation ability of the model. Finally, we fix the readout layers, the imaging modules, and the classification module to the same architecture of GUIDE.

**Classification Performance** Table 5 reports the 10-fold CV testing performance of the baselines. Once again, we repeat the CV 10 times to obtain the standard deviation of each metric. We note that G-MIND with subselection of SNPs performs poorly compared to GUIDE. This may be the result of the complex and polygenic nature of schizophrenia. Finally, we compare GUIDE with two graph convolution-based models. In the first model, we see that replacing the structured representation of the nodes with a random hierarchical graph reduces performance. Hence, the ontological representation is extracting meaningful information from the data. The second model compares GUIDE with a standard GCN and max-pooling. This GCN uses a dense graph for convolution and relies on a data-driven strategy for dimensionality reduction. Again, we observe poor performance, in this case likely due to the large model size. In comparison, GUIDE operates on a sparse graph and relies on biological knowledge for dimensionality reduction on small datasets.

GUIDE shows similar performance with both a random dropout and a BFS layer. Thus, both strategies are effective in regularizing the model. However, the difference becomes apparent when we consider robustness of the selected biomarkers. To this end, we we evaluated the stability of the input importances of GUIDE with random dropout using K-SHAP. We also apply K-SHAP to a version of GUIDE where we remove the dropout layer altogether (i.e., "No Feature Selection" reported in Appendix A.3). In Fig. 10 we see that the reproducibility is lower in both baseline models than the BFS strategy. Hence, our BFS procedure maintains performance while identifying robust biomarkers.

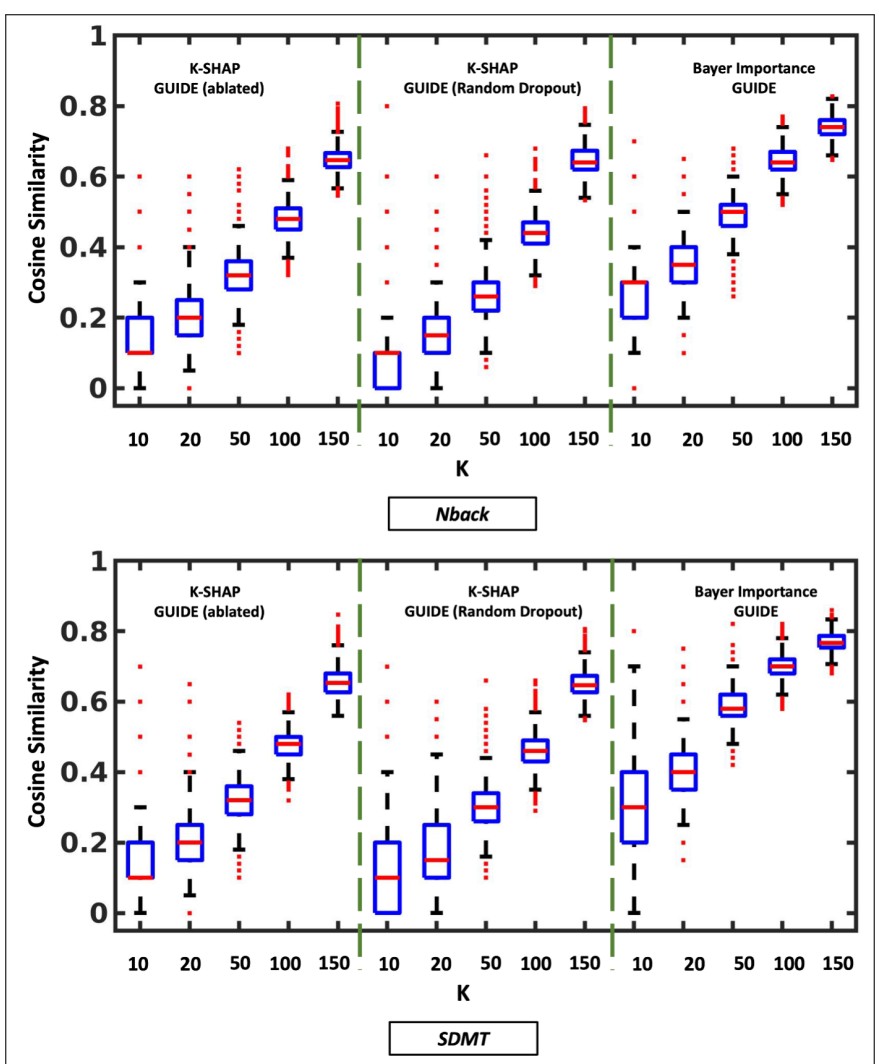

Figure 10: The reproducibility of imaging biomarkers when the input layer of GUIDE is trained without dropout, with random dropout, and with Bayesian feature selection.

