# OpenReview forum: "A Biologically Interpretable Graph Convolutional Network to Link Genetic Risk Pathways and Imaging Phenotypes of Disease "
_ICLR.cc/2022/Conference — ICLR 2022 Poster_

### Official Review · Reviewer_b8wX · 2021-10-25

**Correctness:** 2
**Technical Novelty And Significance:** 3
**Empirical Novelty And Significance:** 2
**Recommendation:** 6
**Confidence:** 4

**Main Review:**

Overall, the paper is well-written and organized.
It is interesting to use the ontological gene hierarchy to define a graph for gene embedding and design imaging features selection in a Bayesian approach. However, the methodological contribution is limited, and several issues need to be justified or supported by experiments.

There should be more rigorous experiments to validate the proposed method, concerning the target classification task, (1) the existing GCN-based methods, (2) the use of preselected SNPs directly, instead of the hierarchical ontology representation, e.g., replacing G-EMBED with G-MIND in GUIDE, (3) random dropout in imaging feature selection in GUIDE.

In the full pipeline of preprocessing and network modeling, none of the test samples be involved in any of the steps in the pipeline.
In this sense, it should be clarified whether the contrast maps generation, regarded as a hyperparameter, was conducted on the training samples only.

The reported performance is relatively too low compared to the existing work on schizophrenia identification with task-/rest-fMRI (mostly higher than 80% in accuracy).

It would be interesting and more informative to illustrate the distribution of cosine similarities for different values of K, instead of a fixed value 50.

It is required to present the validity of handling missing modalities by using public datasets, possibly missing genetic or imaging data.


**Summary Of The Paper:**

This work introduces a framework for fusing genetic and imaging data for schizophrenia classification. In particular, the authors propose to exploit the ontological gene hierarchy to define a graph reflecting genetic risk pathways and to use an attention mechanism for subject-level graph edges identification. They also implemented a Bayesian feature selection strategy using a dropout mechanism in discriminative imaging feature extraction.

**Summary Of The Review:**

While the paper is well organized and written overall, the methodological innovation is limited and the arguments were not well supported from the experiments.

---

> ### Author Response · Authors · 2021-11-19
> **Response for Reviewer b8wX  (1/2)**
>
> **[Novelty]** We thank the reviewer for their assessment. To our knowledge, GUIDE is the first model that incorporates a priori scientific information to regularize the flow of genetic information through a deep neural network. On the imaging front, our Bayesian feature selection procedure extracts a robust and compact set of biomarkers to facilitate interpretability. Thus, we believe our unified framework for whole-brain and whole-genome analysis represents a novel contribution to the imaging-genetics field.
>
> **[Additional Baselines]** We thank the reviewer for suggesting the additional baselines. In our revised manuscript we compare the performance of  GUIDE with G-MIND (after sub-selection of SNPs), GUIDE when replacing the BFS layer with random dropout, a graph convolutional network with hierarchical pooling, and finally a standard graph convolutional network with max-pooling. The results are provided as a separate section in Appendix A.5.
>
> **Table 1: Classification performance (mean $\pm$ std) across repeated CV runs. P-values obtained from DeLong test indicate significantly greater AUROC for GUIDE than each of the baselines**
>
> |    | Sensitivity | Specificity     | Accuracy | AUPRC | AUROC| P-value|
> |    :----:   |:----:   |:----:   |:----:   |:----:   |:----:   | :----:   |
> |GUIDE(Random Dropout) | $0.51 \pm 0.14$ | $ \mathbf{0.79 \pm 0.12}$ | $\underline{0.66  \pm 0.02}$ | $\mathbf{0.70 \pm 0.02}$ | $\mathbf{0.75 \pm 0.01}$ | $0.27$|
> G-MIND~(Sub-selection)  |  $\mathbf{0.63 \pm 0.08}$ | $0.67 \pm 0.06$ | $0.65 \pm 0.01$ | $0.63 \pm 0.04$ | $0.70 \pm  0.02$ | $<10^{-4}$
> Hierarchical GCN |  $0.48 \pm 0.17$ | $0.75 \pm 0.13$ | $0.62 \pm 0.02$ | $\underline{0.65 \pm 0.02}$ | $\underline{0.71 \pm 0.02}$ |  $1.8\times10^{-4}$
> GCN+MaxPooling | $0.43 \pm 0.19$ | $\underline{0.76 \pm 0.14}$ | $0.61 \pm 0.03$ | $0.64 \pm 0.02$ | $ 0.69 \pm 0.02$ |  $<\times10^{-4}$
> GUIDE | $ \underline{0.62 \pm 0.04}$ | $\underline{0.76 \pm 0.04}$ | $\mathbf{0.69 \pm 0.01}$ | $\mathbf{0.70 \pm 0.03}$ | $\mathbf{0.75 \pm 0.01}$
>
>
>
>
> In Table 1 we see a similar performance of GUIDE with random dropout and Bayesian feature selection layer. This shows that both strategies are equally effective in regularizing the model. However, the difference becomes apparent when we consider robustness of the selected biomarkers. Thus, in a subsequent analysis, we evaluated the stability of the input importances of GUIDE with random dropout using K-SHAP. We also apply K-SHAP to a version of GUIDE where we remove the dropout layer altogether (i.e., "No Feature Selection" reported in Appendix A.3). The results are displayed in Fig. 10 of Appendix A.5. As seen, the reproducibility is lower in both baseline models than the Bayesian selection strategy. Hence, our BFS procedure maintains performance while identifying robust biomarkers.
>
> We note that G-MIND with subselection of SNPs performs poorly compared to GUIDE. This may be the result of the complex and polygenic nature of schizophrenia. Finally, we compare GUIDE with two graph convolution-based models. In the first model, we see that replacing the structured representation of the nodes with a random hierarchical graph reduces performance. Hence, the ontological representation is extracting meaningful information from the data. The second model compares GUIDE with a standard GCN and max-pooling. This GCN uses a dense graph for convolution and relies on a data-driven strategy for dimensionality reduction. Again, we observe poor performance, in this case likely due to the large model size. In comparison, GUIDE operates on a sparse graph and relies on biological knowledge for dimensionality reduction on small datasets.

---

> > ### Comment · Reviewer_NHuX · 2021-11-19
> > **Novelty**
> >
> > Just noting that the use of GO etc. as priors for DL architecture has been done before in a number of studies. See "Incorporating biological structure into machine learning models in biomedicine" for a few examples. And another recent study among others here: "Deep GONet: self-explainable deep neural network based on Gene Ontology for phenotype prediction from gene expression data".

---

> > > ### Author Response · Authors · 2021-11-19
> > > **Response to Reviewer NHuX**
> > >
> > > We thank the reviewer for suggesting these papers. We have added them to Section 2 (Related Work) in our latest manuscript revision. With respect to the second paper “Deep GONet: self-explainable deep neural network based on Gene Ontology for phenotype prediction from gene expression data”, we would like to point out that this paper was published in late September 2021, and as such, was developed concurrently with our work.

---

> ### Author Response · Authors · 2021-11-19
> **Response to Reviewer b8wX (2/2)**
>
> **[Contrast Map Generation]** The contrast maps are generated at the subject level (see details about the imaging features in Appendix A.3). We do not perform any secondary thresholding or cluster analysis on the contrast maps, so the training subjects do not influence the generation of contrast maps for the testing subjects.
>
> **[Comparison with fMRI literature]** We thank the reviewer for this comment but politely disagree with the conclusion. A broad search of the schizophrenia literature suggests that, given a similar sample size to this work, most fMRI-based classification models report AUCs and accuracies in the range of 60% -- 80% [1,2,3,4,5]. A large multisite study of schizophrenia [5] also concluded that performances are highly variable across sites and that the higher metrics may be the result of overfitting on site-specific characteristics.
>
> Finally, we emphasize that unlike previous studies, GUIDE allows the user to discover robust and interpretable biomarkers and pathways that link the neuroimaging and genetic modalities with disease phenotype.
>
> **[Cosine Similarities]** We thank the reviewer for suggesting this analysis. We added this comparative study in Appendix A.1.1, Fig. 8.
>
> Through this analysis, we see that the  Bayesian feature selection procedure achieves significantly higher cosine similarity values, which suggests that it selects a more robust set of features that is consistent across subsets of our main cohort.
>
> **[Public Dataset and Missing Modality]** While we certainly appreciate this suggestion, from a practical standpoint, we do not have access to a public imaging-genetics dataset that would be appropriate for GUIDE. First, it is incredibly difficult to gain access to raw SNP information. For example, the Enigma consortium only releases the summary statistics for their GWAS analysis. Second, very few genetics datasets contain parallel neuroimaging. For example, the PGC schizophrenia consortium has sequencing data for thousands of patients, but only a few sites performed MRI scanning. PGC also does not release the raw SNP data. Finally, our motivation is to learn biomarkers of neuropsychiatric illness, which requires data from patient populations, thus adding a layer of complexity to the acquisition. For example, the Human Connectome Project has both imaging and genetics data but consists of only healthy controls.
>
> With that said, we can evaluate the benefit of selectively updating portions of the deep network (i.e., handling missing modalities) through our pICA+RF baseline. As described, we have used a data imputation strategy for the missing modalities in order to maintain a consistent input feature dimensionality. Table 1 shows that this model performs worse than G-MIND and GUIDE, both of which use the selective updating strategy.
>
> **References:**
>
> [1] Koch, S. P., Hägele, C., Haynes, J. D., Heinz, A., Schlagenhauf, F., & Sterzer, P. (2015). Diagnostic Classification of Schizophrenia Patients on the Basis of Regional Reward-Related fMRI Signal Patterns. PLoS ONE, 10(3). https://doi.org/10.1371/JOURNAL.PONE.0119089
>
> [2] Zhang, T., & Davatzikos, C. (2013). Optimally-Discriminative Voxel-Based Morphometry significantly increases the ability to detect group differences in schizophrenia, mild cognitive impairment, and Alzheimer’s disease. NeuroImage, 79, 94–110. https://doi.org/10.1016/J.NEUROIMAGE.2013.04.063
>
> [3] Anderson, A., & Cohen, M. S. (2013). Decreased small-world functional network connectivity and clustering across resting state networks in schizophrenia: An fMRI classification tutorial. Frontiers in Human Neuroscience, 0(SEP), 520. https://doi.org/10.3389/FNHUM.2013.00520/BIBTEX
>
> [4] Anticevic, A., Cole, M. W., Repovs, G., Murray, J. D., Brumbaugh, M. S., Winkler, A. M., Savic, A., Krystal, J. H., Pearlson, G. D., & Glahn, D. C. (2014). Characterizing Thalamo-Cortical Disturbances in Schizophrenia and Bipolar Illness. Cerebral Cortex, 24(12), 3116–3130. https://doi.org/10.1093/CERCOR/BHT165
>
> [5] Orban, P., Dansereau, C., Desbois, L., Mongeau-Pérusse, V., Giguère, C. É., Nguyen, H., Mendrek, A., Stip, E., & Bellec, P. (2018). Multisite generalizability of schizophrenia diagnosis classification based on functional brain connectivity. Schizophrenia Research, 192, 167–171. https://doi.org/10.1016/J.SCHRES.2017.05.027

---

### Official Review · Reviewer_1JN7 · 2021-11-01

**Correctness:** 2
**Technical Novelty And Significance:** 1
**Empirical Novelty And Significance:** 2
**Recommendation:** 5
**Confidence:** 5

**Main Review:**

- Major comments
The paper is very poorly structured with no cohesive approach. For example, the imaging and genetics inputs are discussed way before the actual data is introduced.

How is the ontology network constructed? The explanation is too spare and incomplete.
Given that the key insight of the paper was to use the ontological network, it is important to explain this in detail.

What are the imaging features that go into the algorithm? Are they the t-values or some other measures? It is important to give the exact information, dimensionality of these features.

Are the tSNE clustering surprising given the ontological structure was used to begin with?

- Minor comments
 1. Spell check and several instances of grammatical error

**Summary Of The Paper:**

This paper is the latest in a quest to combine imaging and genetics information for diagnosis and/or biomarker identification. The key idea here is to use the prior knowledge of biological processes as an inductive bias to set up the graph architecture.
The paper also proposes Bayesian approaches to weigh feature importance for biomarker discovery.


**Summary Of The Review:**

The basic idea and premise of the paper are promising. But the quality is presentation is very poor along with insufficient explanations and experiments to justify publication.

---

> ### Author Response · Authors · 2021-11-19
> **Response to Reviewer 1JN7**
>
> **[Presentation of Manuscript]** We politely disagree with the reviewer and believe that we have clearly articulated the problem setup, approach, modeling details, validation procedure, and experimental results. With regards to the data introduction, we chose to provide concrete examples for the imaging and genetic inputs as context for the reader to help them understand how the information is being processed. We also state that the same framework can be used for other data modalities. Finally, if the reviewer has specific suggestions for improvement, we are happy to include these changes in the manuscript.
>
> **[Gene Ontology]** As briefly described in Section 1 (page 2), we rely on the work of [1,2] to construct the ontology. The ontology provides a pre-defined hierarchical network of biological processes, which has been curated by experts in biology. From a modeling perspective, the biological processes can be thought of as nodes in a graph. The two main parts of the ontology in [1,2] are the assignment of genes to these biological processes (i.e., the input projection layer) and the directed edges between the biological processes (i.e., the hierarchical graph).
>
>  Given this comment, we have added a paragraph in Appendix A.3 on gene ontology as follows:
>
> *“The ontology provides a pre-defined hierarchical network of biological processes, which has been curated by experts in biology. From a modeling perspective, the biological processes can be thought of as nodes in a graph. The two main parts of the ontology in [1,2] are the assignment of genes to these biological processes (i.e., the input projection layer) and the directed edges between the biological processes (i.e., the hierarchical graph).”*
>
> **[Imaging Features]** We thank the reviewer; this information has already been provided in Appendix A.2. We will add a sentence in the manuscript to direct the reader to the appendix.
>
> **[Pathway Identification]** The nodes of the ontology are the superset of all biological processes. Among them,  GUIDE automatically learns subject-specific edge weights that tracks pathways (and the associated biological processes) correlated with the disorder. Fig. 6 illustrates the significantly different pathways between patients and controls, as clustered by keywords. To our knowledge, this data-driven pathway identification is the first of its kind. While preliminary, it demonstrates  GUIDE as a novel tool to discover new biomarkers that traditional approaches would not be able to identify. Finally, we emphasize that there is no accepted ground truth in imaging-genetics, so a rigorous validation of the clustering patterns would require new biological experiments.
>
> **[Response to Minor Comment]** We have made the changes corresponding to Reviewer 1 and are happy to correct further typos suggested by any of the reviewers.
>
> **References:**
>
> [1] Ashburner, M., Ball, C. A., Blake, J. A., Botstein, D., Butler, H., Cherry, J. M., Davis, A. P., Dolinski, K., Dwight, S. S., Eppig, J. T., Harris, M. A., Hill, D. P., Issel-Tarver, L., Kasarskis, A., Lewis, S., Matese, J. C., Richardson, J. E., Ringwald, M., Rubin, G. M., & Sherlock, G. (2000). Gene Ontology: tool for the unification of biology. Nature Genetics 2000 25:1, 25(1), 25–29. https://doi.org/10.1038/75556
>
> [2] Mi, H., Muruganujan, A., Casagrande, J. T., & Thomas, P. D. (2013). Large-scale gene function analysis with the PANTHER classification system. Nature Protocols 2013 8:8, 8(8), 1551–1566. https://doi.org/10.1038/nprot.2013.092

---

### Official Review · Reviewer_NHuX · 2021-11-02

**Correctness:** 3
**Technical Novelty And Significance:** 3
**Empirical Novelty And Significance:** 2
**Recommendation:** 8
**Confidence:** 4

**Main Review:**

Strengths
1. The paper is well written overall, with problems of existing methods nicely laid out and provides just the right level of details for intuition.
2. Most existing methods stop at gene-to-geneset when incorporating prior info. GUIDE additionally encodes a hierarchy of relationships between biological processes.
3. The experiments carefully test each property of GUIDE, i.e. compared structured vs. random vs. fully connected graph as well as with and without imaging data.
4. GUIDE achieves better classification performance than some state-of-the-art baselines, and features highlighted by GUIDE match with what are known to be implicated in schizophrenia.

Weaknesses
1. The motivation for this work is partly flawed. While it is true that many existing methods do not consider phenotypes, this is not the intended application of those methods. Instead, those methods are for finding associations between imaging and genetics, which GUIDE does not do, i.e. GUIDE does directly not tell us which genes are associated with which brain regions.
2. How was polygenic score computed? Are cis SNPs of each gene aggregated using univariate GWAS beta? Any pruning of SNPs prior to combining? How was LD accounted for?
3. There are a lot of hyperparameters in the model. Which hyperparameters are important and which do not affect results (as) much? How to set the important ones?
4. In Fig. 2, while prior info helps, the overall AUROC is not particularly high. What is the best reported AUC with comparable genetics data in the literature? For example, in “Artificial image objects for classification of schizophrenia with GWAS-selected SNVs and convolutional neural network”, seems like their proposed method achieved AUC > 0.7 with just genetics. Also, in “Application of deep canonically correlated sparse autoencoder for the classification of schizophrenia states”, accuracy of 95.65% for SNP data sets and an accuracy of 80.53% for fMRI data sets were attained.


**Summary Of The Paper:**

This paper proposes GUIDE for integrating imaging and genetics data for predicting phenotypes. GUIDE uses prior info from gene ontology to restrict connections between genes and biological processes under a graph convolution network (GCN) framework. The genetic and imaging representations are combined for phenotype prediction. Graph pooling and graph attention are used for finding key pathways, and Bayesian feature selection is used for finding key imaging features. GUIDE is evaluated on a schizophrenia dataset.

**Summary Of The Review:**

The paper proposes a useful tool for integrating imaging and genetics with improved accuracy shown on schizophrenia prediction. The proposed interpretability strategy also seems useful. While many parts are based on existing methods, working out the details for combining various components is nontrivial.

Post-revision Summary
While the authors have addressed all my comments, the results are not quite at the level for a score of 10. So I will be keeping my score as 8.

---

> ### Author Response · Authors · 2021-11-19
> **Response to Reviewer NHuX (1/3)**
>
> **[Motivation]** The reviewer makes a good point that the goal of traditional imaging-genetics methods was to relate the biomarkers in each domain. However, there has been a growing awareness that this approach is limited, as the extracted biomarkers may not be representative of the end phenotype (e.g., disease status). This drawback has motivated recent works in imaging-genetics [1,2,3,4] that link the biomarkers with disease status.
>
> While GUIDE does not draw a direct association between brain regions and genetic variation, it is similar in spirit to representation learning techniques used in imaging-genetics (e.g., CCA, pICA) that try to align the data spaces. GUIDE also goes one step further by incorporating the biologically-informed structure of the data into this latent projection operation. Not only does this lead to predictive performance gains (see Table 1), but it also gives us a way to conceptualize the “learning pathways” with respect to the underlying biology. Taken together, we believe that GUIDE is an important contribution to imaging-genetics, and it is well-aligned with the current directions of the field.
>
> **[Genetic Score Calculation]** We apologize for the confusion regarding the polygenic score calculation. We have considered the cis SNPs within 50kb of base pairs to the nearest genes. The gene specific score is calculated as the weighted average of the SNPs using GWAS effect size. We note that the GWAS was performed on a separate SNP dataset that did not include our site. Finally, we have considered 102K LD independent (r2 <0.1) index SNPs for our analysis. We will change our Section 4 on data and preprocessing such that it reads:
>
> *“Genotyping was done using variate Illumina Bead Chipset including 510K/ 610K/660K/2.5M. Quality control and imputation were performed using PLINK and IMPUTE2. The resulting 102K linkage disequilibrium independent (r2 < 0.1) indexed SNPs are grouped to the nearest gene (within 50kb basepairs) (Wong et al., 2017).  The 13,908 dimensional input genes cores are computed as weighted average of the SNPs using GWAS effect size (Ripke et al., 2014). We note that the GWAS was performed on a separate dataset that did not include our site.”*
>
> **References:**
>
>
> [1] Hao, X., Li, C., Du, L., Yao, X., Yan, J., Risacher, S. L., Saykin, A. J., Shen, L., & Zhang, D. (2017). Mining Outcome-relevant Brain Imaging Genetic Associations via Three-way Sparse Canonical Correlation Analysis in Alzheimer’s Disease. Scientific Reports 2017 7:1, 7(1), 1–12. https://doi.org/10.1038/srep44272
>
> [2] Ghosal, S., Chen, Q., Pergola, G., Goldman, A. L., Ulrich, W., Berman, K. F., Blasi, G., Fazio, L., Rampino, A., Bertolino, A., Weinberger, D. R., Mattay, V. S., & Venkataraman, A. (2021). A generative-discriminative framework that integrates imaging, genetic, and diagnosis into coupled low dimensional space. NeuroImage, 238, 118200. https://doi.org/10.1016/J.NEUROIMAGE.2021.118200
>
> [3] Batmanghelich, N. K., Dalca, A., Quon, G., Sabuncu, M., & Golland, P. (2016). Probabilistic Modeling of Imaging, Genetics and Diagnosis. IEEE Transactions on Medical Imaging, 35(7), 1765–1779. https://doi.org/10.1109/TMI.2016.2527784
>
> [4] Yan, J., Risacher, S. L., Nho, K., Saykin, A. J., & Shen, L. (2017). Identification of discriminative imaging proteomics associations in Alzheimer’s disease via a novel sparse correlation model. Pacific Symposium on Biocomputing, 0(212679), 94–104. https://doi.org/10.1142/9789813207813_0010

---

> > ### Comment · Reviewer_NHuX · 2021-11-19
> > **Suggestion for rewording motivation**
> >
> > Incorporating phenotypes into imaging-genetics models is definitely important. My comment is more towards describing existing works more fairly. Instead of describing not including phenotypes as a drawback, which is not the intent of many methods, you could simply say prior works focused on finding associations between imaging and genetics, whereas recent works are starting to additionally incorporate phenotypes. This way, you get the same message across without bending the intentions of prior works.

---

> > > ### Author Response · Authors · 2021-11-19
> > > **Incorporated in the revised manuscript.**
> > >
> > > We thank the reviewer for this suggestion. We have incorporated it in the Introduction (paragraph 2) of our revised manuscript.
> > >
> > > The paragraph now reads:
> > >
> > > *Imaging-genetics is an emerging field that tries to merge these complementary viewpoints (Hariri & Weinberger, 2003). The imaging features are often derived from structural and functional MRI(s/fMRI), and the genetic variants are typically captured by Single Nucleotide Polymorphisms (SNPs). Data-driven imaging-genetics methods can be grouped into four categories. The first category uses sparse multivariate regression to predict one or more imaging features based on a linear combination of SNPs (Wang et al., 2012; Liu et al., 2014).  The second category uses representation learning ( Pearlson et al., 2015; Hu et al., 2018) to maximally align the data distributions of the two modalities. These prior works focus on finding associations between imaging and genetics, whereas recent works are starting to incorporate additional phenotypes like disease status.  The third category relies on generative models (Batmanghelich et al., 2016; Ghosal et al., 2019) to fuse imaging and genetics data with subject diagnosis. While this fusion extracts discriminative biomarkers, the generative nature makes it hard to compensate for missing data or to add data modalities. In fact, clinical neuroscience is moving towards multimodal imaging acquisitions where different modalities contain potentially orthogonal information about the disease.  However, with multimodal data come the problem of missing data acquisitions, thus underscoring the need for flexible and adaptable methods.*

---

> ### Author Response · Authors · 2021-11-19
> **Response to Reviewer NHuX (2/3)**
>
> **[Hyperparameters]** We thank the reviewer for the comment. One advantage of GUIDE is that we have principled ways to set the model architecture and hyperparameters. For example, the architectures for the genetics encoder and decoder are fixed a priori according to the biology literature [1, 2] and would remain constant across datasets and applications.
>
> The user-specified hyperparameters in this work are the latent space dimensionality, the architectures for the classifier and imaging encoder/decoder, and the weights on the loss terms in Eq. (7). To avoid bias, we fix the weight of genetic reconstruction loss, the classification module, and latent space dimension based on our pretraining dataset (i.e., the 1848 genetics-only subjects) and use these values for analysis on our main cohort (i.e., the 208 imaging-genetics subjects). The architectures for our imaging encoder and decoder are derived from the G-MIND model of [3]. Namely, we start with the same two-layer ANN and vary the hidden layer widths over the range 10-100. As we observe just a 2% variation in AUC, we simply fix the width in the middle at 50 hidden nodes per layer. Finally, we fix the loss weight  (\lambda_i) of the imaging loss term as the appropriate power of 10 such that the imaging and genetic losses have similar contributions to the overall loss function.
>
> From a system viewpoint, we believe that the individual modules of GUIDE (hierarchical graph convolution, decoding operation, Bayesian feature selection) play a bigger role in predictive performance than the above hyperparameters. We have included a detailed analysis of the effects of each module in Appendix A.4.
>
>
> **References:**
>
> [1] Ashburner, M., Ball, C. A., Blake, J. A., Botstein, D., Butler, H., Cherry, J. M., Davis, A. P., Dolinski, K., Dwight, S. S., Eppig, J. T., Harris, M. A., Hill, D. P., Issel-Tarver, L., Kasarskis, A., Lewis, S., Matese, J. C., Richardson, J. E., Ringwald, M., Rubin, G. M., & Sherlock, G. (2000). Gene Ontology: tool for the unification of biology. Nature Genetics 2000 25:1, 25(1), 25–29. https://doi.org/10.1038/75556
>
> [2] Mi, H., Muruganujan, A., Casagrande, J. T., & Thomas, P. D. (2013). Large-scale gene function analysis with the PANTHER classification system. Nature Protocols 2013 8:8, 8(8), 1551–1566. https://doi.org/10.1038/nprot.2013.092
>
> [3]Ghosal, S., Chen, Q., Pergola, G., Goldman, A. L., Ulrich, W., Berman, K. F., Blasi, G., Fazio, L., Rampino, A., Bertolino, A., Weinberger, D. R., Mattay, V. S., & Venkataraman, A. (2021). G-MIND: an end-to-end multimodal imaging-genetics framework for biomarker identification and disease classification. In B. A. Landman & I. Išgum (Eds.), Medical Imaging 2021: Image Processing (Vol. 11596, p. 8). SPIE. https://doi.org/10.1117/12.2581127

---

> ### Author Response · Authors · 2021-11-19
> **Response to Reviewer NHuX (3/3)**
>
> **[Comparison with Literature]** We thank the reviewer for suggesting these papers as performance benchmarks.
>
> With respect to the first paper (Xiangning Chen et al., 2021), a careful read will show that the authors achieve similar patient/control AUC (0.72 in Table 2 versus 0.69 for the ontology network in GUIDE). The slight performance variation can be attributed to the limited dataset sizes and possible differences in subject selection and data acquisition. The authors then conduct a secondary analysis, in which they first stratify subjects according to polygenetic risk and classify just those subjects in lowest and highest risk classes. This stratification procedure effectively separates the distributions of patients and controls based on their genetic markers, which is why the secondary analysis achieves much higher AUC. However, it is not a fair comparison with GUIDE, which does not perform SNP sub-selection or subject stratification. Finally, we note that the genetics architecture proposed by ((Xiangning Chen et al., 2021) is not interpretable and does not leverage structural dependencies in genetic data.
>
> With respect to the second paper (Gang Li et al., 2020), we believe the performance is overly optimistic and likely biased. The authors use a single training/testing split and report the performance on just 30 testing subjects rather than run a full K-fold classification. Hence, the model may not generalize to different testing sets. In addition, in Fig. 6 The authors report perfect prediction (accuracy = 1) from SNPs data. Though schizophrenia is a genetic disorder it also has multiple environmental factors [8]. So, perfectly predicting it from genetics data is widely-believed impossible. The reported performance is an extreme outlier compared to the bulk of published work in the fMRI and genetics literature. Therefore, it should be regarded with caution, particularly when used as a benchmark.
>
> Beyond these papers, most fMRI-based classification models for schizophrenia have reported AUCs and accuracies in the range of 60% -- 80% [1,2,3,4,5]. Furthermore, a large fMRI study of schizophrenia [9] has shown that the higher accuracies in this range may be the result of overfitting on site-specific characteristics of the data. Likewise, SNP-based classification models of schizophrenia classification have reported AUC and accuracy performance in the range of 60 -- 80% [6, 7]. Taken together, GUIDE achieves commensurate performance to previous work.
>
> Finally, we emphasize that, unlike previous studies, GUIDE allows the user to discover robust and interpretable biomarkers and pathways that link the neuroimaging and genetic modalities with disease phenotype.
>
> **References:**
>
> [1] Koch, S. P., Hägele, C., Haynes, J. D., Heinz, A., Schlagenhauf, F., & Sterzer, P. (2015). Diagnostic Classification of Schizophrenia Patients on the Basis of Regional Reward-Related fMRI Signal Patterns. PLoS ONE, 10(3). https://doi.org/10.1371/JOURNAL.PONE.0119089
>
> [2] Zhang, T., & Davatzikos, C. (2013). Optimally-Discriminative Voxel-Based Morphometry significantly increases the ability to detect group differences in schizophrenia, mild cognitive impairment, and Alzheimer’s disease. NeuroImage, 79, 94–110. https://doi.org/10.1016/J.NEUROIMAGE.2013.04.063
>
> [3] Anderson, A., & Cohen, M. S. (2013). Decreased small-world functional network connectivity and clustering across resting state networks in schizophrenia: An fMRI classification tutorial. Frontiers in Human Neuroscience, 0(SEP), 520. https://doi.org/10.3389/FNHUM.2013.00520/BIBTEX
>
> [4] Anticevic, A., Cole, M. W., Repovs, G., Murray, J. D., Brumbaugh, M. S., Winkler, A. M., Savic, A., Krystal, J. H., Pearlson, G. D., & Glahn, D. C. (2014). Characterizing Thalamo-Cortical Disturbances in Schizophrenia and Bipolar Illness. Cerebral Cortex, 24(12), 3116–3130. https://doi.org/10.1093/CERCOR/BHT165
>
> [5] Orban, P., Dansereau, C., Desbois, L., Mongeau-Pérusse, V., Giguère, C. É., Nguyen, H., Mendrek, A., Stip, E., & Bellec, P. (2018). Multisite generalizability of schizophrenia diagnosis classification based on functional brain connectivity. Schizophrenia Research, 192, 167–171. https://doi.org/10.1016/J.SCHRES.2017.05.027
>
> [6] van Hilten, A., Kushner, S. A., Kayser, M., Arfan Ikram, M., Adams, H. H. H., Klaver, C. C. W., Niessen, W. J., & Roshchupkin, G. V. (2021). GenNet framework: interpretable deep learning for predicting phenotypes from genetic data. Communications Biology 2021 4:1, 4(1), 1–9. https://doi.org/10.1038/s42003-021-02622-z
>
> [7] Ripke, S., Neale, et al. (2014). Biological insights from 108 schizophrenia-associated genetic loci. Nature 2014 511:7510, 511(7510), 421–427. https://doi.org/10.1038/nature13595
>
> [8] Dean K, Murray RM. Environmental risk factors for psychosis. Dialogues Clin Neurosci. 2005;7(1):69-80. doi: 10.31887/DCNS.2005.7.1/kdean. PMID: 16060597; PMCID: PMC3181718.

---

### Official Review · Reviewer_WH5x · 2021-11-03

**Correctness:** 4
**Technical Novelty And Significance:** 3
**Empirical Novelty And Significance:** 3
**Recommendation:** 6
**Confidence:** 2

**Main Review:**

Minor comments:
Eq. (1): A pair of parentheses needs to be put around $\mathbf{W}_t \otimes \mathbf{A}_g[:,t]$.
Second line below Eq. (1): There is a redundant ``that" in ``that that".
``an binary indicator vector" should be ``a binary indicator vector" (i.e., with article ``a" instead of ``an" before ``binary").
In Section 4, it says ``1848 subjects (793 schizophrenia, 1056 control)". But 793+1056 = 1849 is not equal to 1848.

**Summary Of The Paper:**

This paper proposes a novel end-to-end framework for whole-brain and whole-genome imaging-genetics. The genetics network uses hierarchical graph convolution and pooling operations to embed subject-level data onto a low-dimensional latent space. The imaging network projects multimodal data onto a set of latent embeddings. For interpretability, a Bayesian feature selection strategy is implemented to extract the discriminative imaging biomarkers; these feature weights are optimized alongside the other model parameters. The imaging and genetic embeddings are coupled with a predictor network, to ensure that the learned representations are linked to phenotype. Experiments on a schizophrenia dataset show that the proposed method hasthe better classification performance than state-of-the-art baselines, and the biomarkers identified are reproducible and confirmed to be closely associated with deficits in schizophrenia.

**Summary Of The Review:**

The paper is generally well written, clearly presented, and a pleasure to read. The technique look sound and contributions look solid.

---

> ### Author Response · Authors · 2021-11-19
> **Response to Reviewer WH5x**
>
> We thank the reviewer for their appreciation of our work and for noting the typos. We have added the parentheses in Eq. (1) and fix the wording issues in the subsequent paragraph. In terms of the dataset, we have 792 (rather than 793) schizophrenia patients and 1056 controls for a total of 1848 subjects. We have corrected the numbers accordingly in the manuscript.

---

### Author Response · Authors · 2021-11-19
**Overview of changes made in the revised manuscript.**

We thank the reviewers for their insightful comments and their suggestions. Their recommendations have greatly improved our validation strategy and comparison study with the baselines. Here, we list the set of changes made to the revised manuscript:

**[Description]** We have modified the “Data and Preprocessing” paragraph in Section 4 to better describe gene score generation from genetic data. Additional description about gene ontology is added in Appendix A.2

**[Evaluation of Bayesian Feature Selection]** As recommended by Reviewer 4. We have added a paragraph about the reproducibility of the imaging biomarkers in Appendix A.1.1. Here, we show the superior performance of Bayesian feature selection coupled with GUIDE.

**[Additional Baselines]** Reviewer 4. suggested new baselines to compare against GUIDE. We have added a section in Appendix A.5 where we explain the baselines and report their performance. We observe that GUIDE performs significantly better than the baselines. These additional baselines strengthen our claims and highlight the significance of our model.

**[Minor Changes]** We have fixed the typos suggested by our reviewers and found by us during the review process. Additionally, we made a minor changes in the Introduction and Related Works, as recommended by Reviewer 2 and Reviewer 4.

---

### Decision · Program_Chairs · 2022-01-20

**Decision:**

Accept (Poster)

**Comment:**

In this paper, the authors present a method that combines genetic data (using a hierarchical, graph convolution approach) with imaging data to predict schizophrenia. The reviewers raised several concerns that the authors have addressed. Some of the concerns were relevant to writing, the authors have clarified these points. Another important concern was about the baselines. The authors added other baselines. One of the baselines they added was GUIDE with random dropout, this baseline performs as well as GUIDE, but the authors argue that GUIDE leads to robust features.

I ask the authors to move the other baseline results, specifically the GUIDE with random dropout, and the relevant discussion to the main manuscript, and to consequently temper the discussion of the bayesian feature selection. Currently these additional results are only in the appendix, and not the text. Conditional on this, I recommend acceptance.